# The Nutraceutical Properties of Sumac (*Rhus coriaria* L.) against Gastritis: Antibacterial and Anti-Inflammatory Activities in Gastric Epithelial Cells Infected with *H. pylori*

**DOI:** 10.3390/nu14091757

**Published:** 2022-04-22

**Authors:** Giulia Martinelli, Marco Angarano, Stefano Piazza, Marco Fumagalli, Andrea Magnavacca, Carola Pozzoli, Saba Khalilpour, Mario Dell’Agli, Enrico Sangiovanni

**Affiliations:** 1Department of Pharmacological and Biomolecular Sciences, University of Milan, 20133 Milan, Italy; giulia.martinelli@unimi.it (G.M.); marco.angarano@guest.unimi.it (M.A.); marco.fumagalli3@unimi.it (M.F.); andrea.magnavacca@unimi.it (A.M.); carola.pozzoli@unimi.it (C.P.); skhalilpour@gmail.com (S.K.); mario.dellagli@unimi.it (M.D.); enrico.sangiovanni@unimi.it (E.S.); 2Mucosal Immunology and Biology Research Center, Massachusetts General Hospital, Boston, MA 02115, USA; 3Department of Ophthalmology, Harvard Medical School, Boston, MA 02115, USA

**Keywords:** Sumac, *Rhus coriaria* L., gastritis, *H. pylori*, inflammation, nutraceuticals, botanicals

## Abstract

Sumac (*Rhus coriaria* L.) is a spice and medicinal herb traditionally used in the Mediterranean region and the Middle East. Since we previously demonstrated Sumac biological activity in a model of tumor necrosis factor alpha (TNF-α)-induced skin inflammation, the present work is aimed at further demonstrating a potential role in inflammatory disorders, focusing on gastritis. For this purpose, different polar extracts (water-W, ethanol-water-EW, ethanol-E, ethanol macerated-Em, acetone-Ac, ethylacetate-EtA) were investigated in gastric epithelial cells (GES-1) challenged by TNF-α or *H. pylori* infection. The ethanolic extracts (E, EW, Em) showed the major phenolic contents, correlating with lower half maximal inhibitory concentrations (IC_50_s) on the release of interleukin-8 (IL-8, <15 μg/mL) and interleukin-6 (IL-6, <20 μg/mL) induced by TNF-α. Similarly, they inhibited IL-8 release (IC_50_s < 70 μg/mL) during *Helicobacter pylori* (*H. pylori*) infection and exhibited a direct antibacterial activity at comparable concentrations (minimum inhibitory concentration (MIC) = 100 μg/mL). The phenolic content and the bioactivity of EW were maintained after simulated gastric digestion and were associated with nuclear factor kappa B (NF-κB) impairment, considered the main putative anti-inflammatory mechanism. On the contrary, an anti-urease activity was excluded. To the best of our knowledge, this is the first demonstration of the potential role of Sumac as a nutraceutical useful in *H. pylori*-related gastritis.

## 1. Introduction

*Rhus coriaria* L. (Sumac) belongs to the Anacardiaceae family, widely grown throughout the Mediterranean region. Leaves and fruits have a remarkable medicinal value in Middle Eastern herbal medicine [1]. The brown/red fruits of *Rhus coriaria* are used as a very popular spice in food production for their sour lemony taste. Phytochemical characterization of berries showed the occurrence of different antioxidants belonging to several classes of polyphenols, among which flavonoids and gallotannins are the most abundant [2]. 

Red fruits are traditionally used in Persian medicine to treat diarrhea, hemorrhoids, gout, and decrease cholesterol, uric acid, and blood sugar levels, and for a variety of other biological activities recently revised by Elagbar et al. [1]. Much evidence supports the pharmacological and nutraceutical properties of Sumac in a variety of diseases [3,4,5,6,7,8]. Moreover, additional studies demonstrated interesting effects in inflammatory conditions, including the reduction of pro-inflammatory mediators and the inhibition of the nuclear factor kappa B (NF-κB) activation in human keratinocytes challenged with tumor necrosis factor alpha (TNF-α) [5].

Gastritis is considered an inflammatory-based condition, which is provoked by several risk factors that include stress, alcohol abuse, the use of drugs such as non-steroidal anti-inflammatory drugs (NSAIDs), and bile reflux.

*Helicobacter pylori* (*H. pylori*) is a Gram-negative bacterium infecting around 50% of people all over the world, the bacterium can cause acute or chronic gastritis or digestive problems [9].

Infection with *H. pylori* leads to the development of more severe diseases, such as a peptic ulcer or gastric cancer. The World Health Organization (WHO) assessed *H. pylori* as a class I carcinogen for gastric cancer in 1994. During *H. pylori* infection, a variety of pro-inflammatory mediators are released by gastric epithelial cells and macrophages, above all cytokines and chemokines (i.e., interleukin-6 (IL-6), TNF-α, and interleukin-8 (IL-8)) downstream of the transcription factor NF-κB.

NF-κB orchestrates the expression and release of IL-8 and IL-6 which, in turn, augment the gastric phlogistic processes [10,11]. Of note, the expression of both IL-8 and IL-6 genes is regulated by NF-κB activation following *H. pylori*-related inflammatory processes [12,13,14,15,16].

*H. pylori* eradication is an important strategy for managing gastric inflammatory conditions; however, combination therapy with several antibacterial agents is limited by the increase of resistance of *H. pylori* strains, often leading to unsuccessful treatment and worsening the disruption of intestinal microbiota; thus, the search for novel antibacterial agents with high selectivity against *H. pylori* strains is a good strategy to preserve intestinal microflora. In recent years, several extracts or isolated compounds from natural sources, or synthetic drugs have been investigated as potential anti-*H. pylori* agents [17].

According to the literature, Sumac has been traditionally used to counteract inflammatory conditions including gastrointestinal ailments; however, no conclusive studies on the effect of *Rhus coriaria* extracts against gastric inflammatory conditions have been reported so far.

The aim of the present work is to assess the potential anti-inflammatory and anti-bacterial activity of *Rhus coriaria* extracts in human gastric epithelial cells challenged with *H. pylori* or TNF-α. Since Sumac fruits contain several classes of constituents with different polarity, another purpose of the study was to identify the most suitable extract type in terms of biological activity and future commercialization as food supplement ingredient. 

Among the six tested extracts, the ethanol-water extract (EW) showed promising activity as anti-inflammatory and anti-*H. pylori* agent, also when subjected to an in vitro mimicked gastric digestion, thus suggesting the beneficial properties of Sumac towards the gastric mucosa and its possible use as an ingredient of food supplements able to prevent inflammatory-based gastric diseases induced by *H. pylori* infection, also limiting bacterial growth.

## 2. Materials and Methods

### 2.1. Extraction Method and Extracts Characterization

All the extracts tested in this study were prepared according to Khalilpour et al. [5]. The extracts were named as follows: 100% water (W), 100% ethanol (E), ethanol–water (ethanol:water 50:50 *v*/*v*, EW), ethanol macerated (plant material subjected to maceration with pure ethanol for 48 h, Em), acetone (Ac), and ethylacetate (EtA). 

The total phenol content, measured as gallic acid equivalents/g of extract, showed a significant amount of polyphenols (above 200 mg/g) in E, EW, Em, and Ac, whereas the amount for W and EtA was found significantly lower (under 100 mg/g). In general, the presence of ethanol as a solvent increased the total phenol content, which was maximum following maceration (Table 1). 

The preliminary characterization through high-performance liquid chromatography ultraviolet/with diode-array detection (HPLC-UV/DAD) analysis, previously published by our group, showed the presence of a significant amount of flavonoids, tannins, and anthocyanins in the EW and Em extracts. Both extracts showed similar amounts of flavonoids (0.23%, flavonoids expressed as quercetin-3-*O*-glucoside) and tannins (4.54% and 4.33%, tannins expressed as gallic acid, for EW and Em, respectively), while EW showed a significantly higher anthocyanin content (0.207%, anthocyanins expressed as cyanidin-3-*O*-glucoside) than the Em extract (0.031%, anthocyanins expressed as cyanidin-3-*O*-glucoside) [5].

### 2.2. Total Phenol Content Assay

Total polyphenol content was measured by Folin–Ciocâlteu’s method. Briefly, the extracts were dissolved in water (5 mg/mL), then 20 μL were diluted to a final volume of 800 μL, corresponding to 100 μg of extracts weight. Then, 50 μL of 2 N Folin–Ciocâlteu reagent (Merck Life Science, Milan, Italy) and 150 μL of 20% (*w*/*v*) sodium carbonate (Na_2_CO_3_) were added. After 30 min of incubation at 37 °C, the absorbance of the samples was measured with a Jasco V630 Spectrophotometer (JASCO International Co. Ltd., Tokyo, Japan) at 765 nm. The total phenol content was measured using a calibration curve of gallic acid. Results were expressed as mg of gallic acid equivalents per g of extract.

### 2.3. Cell Culture

Gastric epithelial cells (GES-1) are human normal gastric epithelial cells, provided with the permission of Dr. Dawit Kidane-Mulat (University of Texas, Austin, TX, USA). GES-1 were cultivated in Roswell Park Memorial Institute Medium (RPMI) 1640 medium (Gibco, Thermo Fisher Scientific, Waltham, MA, USA), added with penicillin 100 units/mL, streptomycin 100 mg/mL, L-glutamine 2 mM (Gibco, Thermo Fisher Scientific, Waltham, MA, USA) and 10% heat-inactivated fetal bovine serum (Euroclone S.p.A, Pero, Italy). Cells were incubated at 37 °C, 5% carbon dioxide (CO_2_), in humidified atmosphere. Cells were detached from the flask every 48–72 h upon reaching confluency (Primo^®^, Euroclone S.p.A., Pero, Italy) by trypsin-ethylenediaminetetraacetic acid (EDTA) 0.25% solution (Gibco, Thermo Fisher Scientific, Waltham, MA, USA), then counted, and seeded in a new flask (1 × 10^6^ cells) for the following sub-culture.

### 2.4. Bacterial Culture

*H. pylori* strain 26695 (KE26695) (ATCC, American Type Culture Collection) was cultured in Petri dishes (Primo^®^, Euroclone S.p.A., Pero, Italy) containing Müller-Hinton Broth (Difco™, BD, Franklin Lakes, NJ, USA) medium supplemented with agar (Merck Life Science, Milan, Italy) and 5% defibrinated sheep blood (TCS Bioscience Ltd., Oxoid, Hampshire, UK). The bacteria were cultivated by solid culture on the blood-agar dishes for 72 h under microaerophilic atmosphere (5% oxygen (O_2_), 10% CO_2_ and 85% nitrogen (N_2_) at 37 °C with 100% humidity). Before each GES-1 infection procedure, the bacterium was recovered from the Petri dishes and the bacterial count was estimated by optical density at 600 nm (O.D. value = 5 corresponds to 2 × 10^8^ bacteria).

### 2.5. Cell Treatment

To measure the release of the pro-inflammatory cytokines and the activation of NF-κB, cells were seeded in 24- or 6-well-plates (Falcon^®^, Corning Life Science, Amsterdam, The Netherlands) at a density of 3 × 10^5^ cells/well and 10^5^ cells/well, respectively. After 72 h, GES-1 were treated with the pro-inflammatory stimulus TNF-α (10 ng/mL) or the bacterium *H. pylori* (bacterium:cell ratio of 50:1) along with the extracts at different concentrations. In the co-treatment with the bacterium, serum starvation was performed using 0.5% serum medium, added with 1% L-glutamine and 1% penicillin/streptomycin, 24 h before. Regarding *H. pylori* infection, all the treatments were conducted with serum and antibiotic-free medium in the co-culture with, while TNF-α treatments were conducted with serum-free medium. During the treatment, cells were maintained in incubator at 37 °C and 5% CO_2_. After 6 h for the release of the pro-inflammatory cytokines or 1 h for NF-κB activity, culture media or cell lysates were collected for biological assessments.

### 2.6. Cytotoxicity Assay

The correct cell morphology was verified by light microscope inspection before and after treatment. Cell viability was assessed by the 3-(4,5-dimethylthiazol-2-yl)-2-5-diphenyltetrazolium bromide (MTT) method (Merck Life Science, Milan, Italy) at the end of the treatments (6 h) [18]. This method is an undirect index of viability, since it evaluates the activity of a mitochondrial enzyme, the succinate dehydrogenase. Briefly, the medium was discarded, then 200 μL of MTT solution (0.1 mg/mL, phosphate buffered saline (PBS) 1X) were added to each well (45 min, 37 °C) and kept in darkness. Then, MTT solution was discarded and the purple salt included into the cells was dissolved by isopropanol:dimethyl sulfoxide (DMSO) (90:10 *v*/*v*), and the absorbance was read at 595 nm (Victor^TM^ X3, Perkin Elmer, Walthman, MA, USA).

### 2.7. Measurement of IL-8 and IL-6 Release

The pro-inflammatory mediators IL-8 and IL-6 were quantified in cell media after 6-h treatments with TNF-α or *H. pylori* as stimuli and the extracts, by an enzyme-linked immunosorbent assay (ELISA), using two sandwich ELISA kits: Human Interleukin-8 ELISA Development Kit and Human Interleukin-6 ELISA Development Kit (Peprotech, London, UK), according to Nwakiban et al. [19] and manufacturer instructions. Briefly, clear plates (Corning enzyme immunoassay/radioimmunoassay (EIA/RIA) plates, 96-well, Merck Life Science, Milan, Italy) were coated with the capture antibody from the ELISA kit (overnight, room temperature (r.t.)). The non-specific binding sites were blocked with albumin 1% for 1 h and then a total of 100 μL of samples in duplicate were transferred into wells at room temperature for 2 h. The pg/mL of IL-8 and IL-6 was detected through the colorimetric reaction due to horseradish peroxidase (HRP)-conjugated biotinylated antibody and 3,3′,5,5′-tetramethylbenzidine (TMB) substrate (Merck Life Science, Milan, Italy). The absorbance was obtained at 450 nm 0.1 s by multiplate reader (Victor^TM^ X3, PerkinElmer, Waltham, MA, USA). Data were expressed as percentage relative to stimulated control, which was arbitrarily assigned the value of 100%.

### 2.8. NF-κB Activation

The activation of NF-κB was measured by western blot and immunofluorescence techniques. Western blot was employed to measure the activation of phospho-p65 in the GES-1 cells, while immunofluorescence was employed to reveal the translocation of p65 in the nucleus of GES-1 cells.

#### 2.8.1. Western Blotting Analysis

Total protein extracts from GES-1 cells were obtained by lysing cells with 200 μL radioimmunoprecipitation assay (RIPA) buffer containing a mix of protease (Protease Inhibitor Cocktail; Merck Life Science, Milan, Italy) and phosphatase inhibitors (1 mM sodium orthovanadate (Na_3_VO_4_) and 5 mM sodium fluoride (NaF)). The protein concentration for each cell lysate was assessed with the bicinchoninic acid (BCA) protein assay method (Euroclone S.p.A., Pero, Italy) and 20 μg of each sample was prepared with 4× Laemmli sample buffer, boiled at 95 °C for 5 min, centrifuged at 16,000× *g* for 1 min, and loaded on sodium dodecyl-sulfate (SDS)-polyacrilamide gel. After gel run, proteins were transferred to a nitrocellulose membrane using the iBlot™ Gel Transfer Device (Invitrogen™, Thermo Fisher Scientific, Waltham, MA, USA) and blocked in tris buffered saline with Tween (TBS-T) containing 5% Bovine Serum Albumine (BSA) (Merck Life Science, Milan, Italy) for 1 h at room temperature. Membranes were incubated overnight at 4 °C with the primary antibodies anti-phospho-p65 (Phospho NF-κB p65 (Ser536) (93H1) Rabbit mAb #3033; Cell Signaling Technology, Danvers, MA, USA) and anti-β-actin (Monoclonal Anti-β-Actin Clone AC-15 produced in mouse; Merck Life Science, Milan, Italy), used as control. Then, after washing with TBS-T solution, membranes were incubated with anti-rabbit and anti-mouse secondary antibodies (Merck Life Science, Milan, Italy), respectively, for 1.5 h at room temperature. Immunocomplexes were visualized with electrochemiluminescence (ECL) (Westar Antares, Cyanagen Srl, Bologna, Italy) and the Chemidoc MP imaging system (Bio-Rad Laboratories Srl, Irvine, CA, USA). Protein levels were quantified with ImageLab 6.1 software (Bio-Rad Laboratories Srl, Irvine, CA, USA). Primary antibodies were diluted as follows: anti-phospho-NF-κB p65 1:1000 *v*/*v*; anti-β-actin 1:2500 *v*/*v*.

#### 2.8.2. Immunofluorescence

Immunofluorescence technique was used to assess the translocation of NF-κB from cytoplasm to nucleus of GES-1 cells, challenged with *H. pylori* and treated with extracts. Cells were seeded on coverslips placed in 24-well plates at the density of 30,000 cells/well. Before treatment *H. pylori* was stained with carboxyfluorescein succinimidyl ester (CFSE) 5 mM (CellTrace™, Cell Proliferation kits; Invitrogen, Thermo Fisher Scientific, Waltham, MA, USA) diluted 1:500 *v*/*v* and incubated for 20 min at 37 °C. Subsequently, fetal bovine serum (FBS) was added to the bacterial suspension to bind the excess of CFSE, followed by three washes with PBS and centrifugation at 6000× *g* for 5 min to remove the excess of CFSE not bound to the bacterium. After 1 h treatment, co-cultures were washed (PBS 1X) and fixed with 4% formaldehyde solution for 15 min at r.t. A 5% BSA blocking solution was added to the well and incubated at room temperature for 1 h. Cells were incubated with the primary antibody (NF-κB p65 (D14E12) XP^®^ Rabbit mAb #8242, Cell Signaling Technology, Danvers, MA, USA) diluted 1:400 *v*/*v* overnight at 4 °C and then with the secondary antibody (Alexa Fluor 647 conjugated with anti-rabbit immunoglobulin G (IgG) (heavy + light (H + L)), F(ab’)2 Fragment #4414, Cell Signaling Technology, Danvers, MA, USA) diluted 1:1000 *v*/*v*. After 2 h, coverslips were washed with PBS and mounted on slides with a drop of ProLong Gold Antifade Reagent with 4′,6-diamidino-2-phenylindole (DAPI) (#8961, Cell Signaling Technology, Danvers, MA, USA), and then were imaged with a confocal laser scanning microscope (LSM 900, Zeiss, Oberkochen, Germany).

### 2.9. Minimum Inhibitory Concentration (MIC)

The microbroth dilution method was performed as per the recommendations of Clinical and Laboratory Standards Institute (CLSI) [20] and was used to determine MIC. Extracts at different concentration and positive control (tetracycline 0.125 μg/mL) were prepared in Brucella broth (BBL™, BD, Franklin Lakes, NJ, USA) supplemented with 5% FBS, and 100 μL of each sample were placed in a 96-well U-bottom plate (Greiner Bio-One™, Rome, Italy). Then, 100 μL of *H. pylori* suspension prepared in the same medium in a dilution adjusted to O.D. value = 0.1 were added to each well. After well-mixing, the 96-well plate was incubated at 37 °C in a 5% CO_2_ incubator under microaerophilic condition. After 72 h, the assay plate was read visually for growth inhibition at 600 nm 0.1 s, using a multi-detection microplate reader (Victor™ X3, PerkinElmer, Waltham, MA, USA).

### 2.10. Urease Activity

The urease activity of *H. pylori* was measured using a phenol red solution (9.1 g/L potassium dihydrogen phosphate (KH_2_PO_4_), 9.5 g/L disodium hydrogen phosphate (Na_2_HPO_4_), 0.01 g/L phenol red) and the substrate urea (2 g/L), according to the protocol used by Svane et al. and Korona-Glowniak et al. [21,22]. In the presence of the bacterium, urea was converted to ammonia (NH_3_), modifying the pH and changing the color of the solution. In a 96-well plate 100 μL of extracts and positive controls were prepared, then *H. pylori* suspension (final concentration O.D. value = 0.1) and 100 μL of phenol red/urea solution were added. After 1 h of incubation at 37 °C, the absorbance was measured with a spectrophotometer at 570 nm 0.1 s (EnVision Plate reader, Perkin Elmer, Waltham, MA, USA).

### 2.11. In Vitro Simulated Gastric Digestion

The gastric digestion was mimicked by an in vitro simulation, as previously described [23]. In brief, EW extract (100 mg) was incubated for 5 min at 37 °C with simulated saliva (6 mL), then gastric juice (12 mL) was added, and the sample was incubated for 2 h at 37 °C. Then, the suspension was centrifuged (5 min, 3000 g) and the supernatant was freeze-dried.

### 2.12. Statistical Analysis

All data were expressed as mean ± standard deviation (SD) of at least three independent experiments; the interval of confidence related to the half maximal inhibitory concentrations (IC_50_s) calculation (% confidence interval (C.I.)) was reported in the Results session.Data were elaborated by unpaired ANOVA test and Bonferroni post-hoc analysis. Statistical measures were conducted by GraphPad Prism 8.0 software (GraphPad Software Inc., San Diego, CA, USA). Values of *p* < 0.05 were considered statistically significant. The IC_50_s were calculated by GraphPad Prism 8.0 software.

## 3. Results

### 3.1. Cytotoxicity of the Extracts in Human Gastric GES-1 Cells

All the extracts were tested for cytotoxicity (MTT assay) in GES-1 cells, incubating increasing concentrations of extracts in presence of TNF-α (Appendix A) or *H. pylori* (Appendix A) for six hours. In both cases, all the extracts did not show any cytotoxicity in the range 10–200 μg/mL. Concentrations used for the biological assays were chosen accordingly. The MTT assay was also performed on the EW extract subjected to in vitro simulated gastric digestion (EWd), at concentrations ranging between 10 and 200 μg/mL. No cytotoxicity was detected in cells treated with each concentration tested in the presence of TNF-α or *H. pylori* as pro-inflammatory stimuli (Appendix A).

### 3.2. Effect of the Extracts on the TNF-α-Induced IL-8 and IL-6 Release in GES-1 Cells

Several cytokines are involved in gastric inflammation including IL-8, IL-6, and TNF-α. IL-8 and IL-6 are produced by gastric epithelial cells following inflammation [10]. The release of TNF-α by gastric epithelial cells and macrophages feeds the inflammatory process in the gastric mucosa, activating the cytokine cascade. To investigate the activity of *Rhus coriaria* in modulating the inflammatory process, GES-1 cells were treated for six hours with the extracts and TNF-α, then IL-8 or IL-6 release was measured through ELISA assay.

All the extracts showed a concentration-dependent inhibition of the TNF-α-induced IL-8 release; the aqueous (W), ethanolic (E), or hydroethanolic (EW) extracts showed the highest inhibitory activity (IC_50_ ranging from 12.1 to 14.7 μg/mL, Table 2) whereas the activity of the acetonic (Ac) and ethylacetate (EtA) ones was lower (IC_50_ ~30 μg/mL, Table 2). For all the extracts, the first concentration which showed statistically significant inhibition of IL-8 release was 10 μg/mL (Figure 1).

IL-6 is known to require NF-κB activation during *H. pylori*-related gastritis. Thus, the potential ability of the extracts to control gastric inflammation acting on IL-6 release was further verified. Likewise, the results on the IL-8 release, all the extracts inhibited the TNF-α-induced IL-6 release in a concentration-dependent fashion (Figure 2).

The IC_50_ values paralleled those obtained on the IL-8 release, with the aqueous, ethanolic, or hydroethanolic extracts showing the most potent effect (IC_50_s ranging from 10.3 to 19.3 μg/mL, Table 2) compared to the acetonic and ethylacetate ones (IC_50_: 69.9 and 85.4 μg/mL, respectively; Table 2). Similar effects were obtained by the aqueous and ethanolic extracts in inhibiting IL-8 and IL-6.

### 3.3. Effect of the Extracts on the H. pylori-Induced IL-8 and IL-6 Release in GES-1 Cells

According to the literature, expression of both the NF-κB dependent IL-8 and IL-6 genes occurs during *H. pylori*-induced gastric inflammation [12,13,14,15,16]. Thus, cells were treated with *H. pylori* and the extracts (50–200 μg/mL), as referred to by the Materials and Methods section, and release of IL-8 or IL-6 was assessed through an ELISA assay. All the extracts were able to affect IL-8 release to a different extent (Figure 3), with IC_50_s ranging between 62.0 and 183.1 μg/mL (Table 3).

The highest activity was found for the ethanolic or hydroethanolic extracts compared to the water extract. On the opposite, all the extracts showed the same negligible activity against IL-6 release, with IC_50_s above 200 μg/mL (Table 3).

### 3.4. Effect of the Extracts on the H. pylori Growth

The exploration of novel compounds with antibacterial activity and a favorable safety profile, based on long-standing consumption as a food, is an attractive therapeutic approach to prevent or ameliorate several gastric inflammatory conditions.

Thus, the effect of the extracts (25–400 μg/mL) on the *H. pylori* growth after 72 h of incubation was investigated. All the extracts, except for the aqueous one, showed a concentration-dependent inhibition of the bacterial growth (Figure 4), with a MIC value of 100 μg/mL for all the active extracts; however, EW and Em showed to better impair bacterial growth since it was completely abolished at 100 μg/mL.

### 3.5. Effect of the In Vitro Digested Hydroethanolic Extract (EWd) on the TNF-α- or H. pylori-induced IL-8 Release in GES-1 Cells

In the next steps, we focused on the EW extract, the choice of which was based on the following experimental indications: the extract showed a high amount of phenols and was among the most active extracts able to inhibit IL-8 release, induced by both *H. pylori* or TNF-α; it was also active in inhibiting *H. pylori* growth and the release of IL-6; it is reasonable to consider the combination of water and ethanol as a safer and cheaper extraction solvent than ethanol alone. 

Thus, the extract was firstly subjected to in vitro simulated gastric digestion with the purpose of assessing its stability in the gastric environment, then total phenol content and IL-8 release induced by *H. pylori* or TNF-α were assessed.

The EW extract subjected to digestion (EWd) showed no decrease in the phenol content compared to the undigested extract (251.8 vs. 258.5 μg/mL, respectively); moreover, once again the extract inhibited both the TNF-α- and *H. pylori*-induced IL-8 release (Figure 5), with low IC_50_s (7.73 and 43.92 μg/mL, respectively).

### 3.6. EW and EWd Extracts Impair the NF-κB Pathway in GES-1 Cells

It has been extensively reported in the literature that the activation of the NF-κB pathway is able to promote the release of IL-8 and IL-6, which in turn lead to the amplification of the gastric inflammatory process [10,11]. Since the EW extract showed interesting effects as an inhibitor of the release of these cytokines we decided to investigate if the NF-κB pathway could be affected by extracts as well, through western blot and immunofluorescence assays. EW and EWd extracts (200 μg/mL) impaired the NF-κB pathway, as evident from the reduction of p65 nuclear accumulation (Figure 6) and phosphorylation at the upstream level (Figure 7). Accordingly, EW extract also reduced the TNF-α-induced NF-κB driven transcription in GES-1 cells at lower concentrations (IC_50_ 50.16 μg/mL, data not shown).

### 3.7. Effect of EWd on H. pylori Growth and Urease Activity

To investigate if the gastric digestion could influence the EW extract activity, the bacterium was incubated for 72 h with increasing concentrations of the EWd extract (25–400 μg/mL), and the growth of *H. pylori* was assessed. The extract elicited a concentration-dependent inhibition of the bacterial growth (Figure 8A) with a MIC of 100 μg/mL, the same of the non-digested extract, suggesting that the simulated gastric digestion did not affect the activity; however, both the extracts did not affect the urease activity at 400 μg/mL (Figure 8B).

## 4. Discussion

*Rhus coriaria* L. (Sumac) fruits are widely used as a spice; previous phytochemical characterizations showed the presence of several antioxidants including flavonoids and gallotannins [5].

A few papers have reported anti-inflammatory activities of Sumac in a variety of diseases. Ahmad et al. reported the anti-ulcer effect of a hydroalcoholic extract (145 and 248 mg/kg) from Sumac in rodent models of stress, ethanol, and indomethacin-induced gastritis [24,25,26]; however, the anti-inflammatory activities in *H. pylori*-related gastritis, as well as the direct effect on *H. pylori* growth, deserved further investigation. In the present study, we aimed at assessing which different *Rhus coriaria* L. fruit extracts, obtained with various solvents, could be the best candidate to be used as an ingredient of food supplements intended as adjuvants against gastric inflammatory conditions.

All the extracts, except for the aqueous one, showed anti-*H. pylori* activity with similar MIC, regardless of the solvent used for extraction. Since the MIC values correlated with the phenolic content (lower in aqueous and ethylacetate extract), we suppose that the efficiency of polyphenol extraction, more than the presence of specific polyphenols, may account for the antibacterial activity. For this reason, the attribution of the antibacterial effect to specific compounds or their combination and the related mechanism of action may deserve dedicated studies. Our results are in line with another work present in the literature, in which the direct anti-*H. pylori* effect of ethanol extract from *Rhus coriaria* showed comparable MIC value (214.28 μg/mL) [27]. Of note, our experiments were conducted using a well-characterized bacterial strain (ATCC 26695), while the mentioned work assessed the antibacterial activity on clinical strains, thus limiting the comparability of results. Another *Rhus* species (*Rhus typhina* L.), widely used in China to treat gastrointestinal disorders, has been previously reported to impair *H. pylori* growth in vitro [28], thus suggesting that antibacterial compounds could be common into the *Rhus* genus. Hydro-alcoholic extracts of *Rhus coriaria* L. showed antibacterial activity against a variety of Gram-positive and Gram-negative food-borne bacteria, but the anti-*H. pylori* effect was not investigated [29].

This is the first study assessing the anti-*H. pylori* potential of different Sumac extracts made up with different solvents with a broad range of polarity; the study shows that the hydroethanolic extract maintains a similar MIC value after in vitro simulated gastric digestion.

Although the EW extract impaired *H. pylori* growth, both EW extracts, non-digested and subjected to simulated gastric digestion, did not influence in vitro urease activity in our experimental conditions. These results seem to be in contrast with the study by Mahernia et al. [30] in which a hydroalcoholic extract from *Rhus coriaria* L. fruits showed an anti-urease activity, with an IC_50_ value of 80.29 μg/mL; however, the different extract used in the study (methanol:water 80:20 *v*/*v*) as well as the different *H. pylori* strain may account for this discrepancy.

*Rhus coriaria* extracts show a strong anti-inflammatory activity, inhibiting IL-8 release induced by *H. pylori* or TNF-α. The effect on IL-8 release was more pronounced when cells were challenged with TNF-α (range: 14.1–32.5 μg/mL); a similar trend was recorded for IL-6 release. This may be due to the complexity of pathways modulated by *H. pylori*, with respect to TNF-α alone, activating a variety of factors including NF-κB and activator protein 1 (AP-1). 

Since during *H. pylori*-induced gastric inflammation there is a massive release of TNF-α from infiltrated immune cells, which activates and prolongs the inflammatory cascade, it is likely that these effects may add up in vivo. NF-κB plays a key role in the release of several cytokines including IL-8 and IL-6 which, in turn, lead to the amplification of the gastric inflammatory processes; thus, we selected NF-κB signalling as a relevant target to assess the anti-inflammatory properties of EW and the matched digested extracts. EW extract was able to impair the NF-κB pathway during TNF-α induction. Accordingly, immunofluorescence showed the inhibition of the *H. pylori*-induced NF-κB (p65) translocation from the cytoplasm into the nucleus for EW and EWd extracts, which was in line with the inhibition of p65 phosphorylation at the upstream level observed in western blots. Consequently, our data suggest that polyphenols from *Rhus coriaria* L. may exert anti-inflammatory effect by interfering with NF-κB at gastric level. 

## 5. Conclusions

In conclusion, the antibacterial and anti-inflammatory effects of the hydroalcoholic extract, which is maintained also when subjected to in vitro simulated gastric digestion, suggest the beneficial properties of Sumac towards inflammation of the gastric epithelium and its possible use as ingredient of food supplements able to prevent inflammatory-based gastric diseases induced by *H. pylori*; the extract may be also able to limit bacterial growth by acting directly against the bacterium. In addition, taking into consideration the plausible dietary intake of grams of fruits and the extraction yields, the bioactivity was observed at concentrations (101–102 μg/mL) easily achievable following oral consumption of the fruits or the corresponding enriched polar extracts. 

Future studies will be devoted to better clarify the role of specific classes of polyphenols present in *Rhus coriaria* L. fruits to recommend standardized products with potential anti-gastritis effect in vivo.

## Figures and Tables

**Figure 1 nutrients-14-01757-f001:**
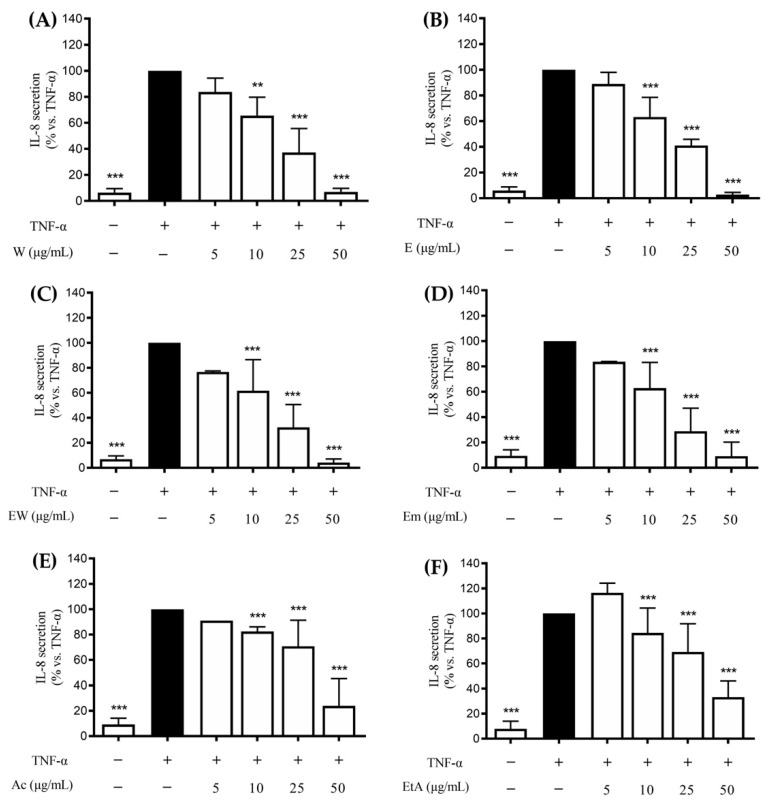
Effects of *Rhus coriaria* L. extracts ((**A**) 100% water (W), (**B**) 100% ethanol (E), (**C**) ethanol–water (ethanol:water 50:50 *v*/*v*, EW), (**D**) ethanol macerated (plant material subjected to maceration with pure ethanol for 48 h, Em), (**E**) acetone (Ac), (**F**) ethylacetate (EtA)) on the TNF-α (10 ng/mL) challenged GES-1 cells (6 h). The release of IL-8 was assessed through an ELISA assay. The reference compound (black color) used was EGCG 20 μM (−37%). Data are reported as percentage in comparison to the stimulated control, which was arbitrarily assigned to 100% value. ** *p* < 0.01, *** *p* < 0.001 versus TNF-α. IL-8, interleukin-8; TNF-α, tumor necrosis factor alpha; GES-1, gastric epithelial cells; ELISA, enzyme-linked immunosorbent assay; EGCG, epigallocatechin gallate; − and +, absence or presence of the respective conditions.

**Figure 2 nutrients-14-01757-f002:**
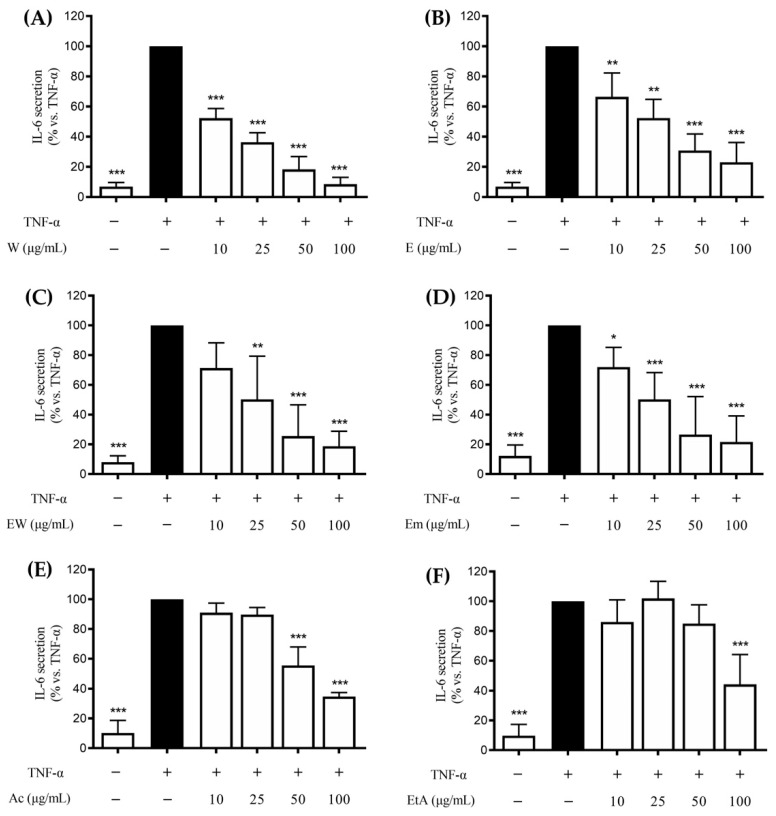
Effects of *Rhus coriaria* L. extracts ((**A**) 100% water (W), (**B**) 100% ethanol (E), (**C**) ethanol–water (ethanol:water 50:50 *v*/*v*, EW), (**D**) ethanol macerated (plant material subjected to maceration with pure ethanol for 48 h, Em), (**E**) acetone (Ac), (**F**) ethylacetate (EtA)) on the TNF-α (10 ng/mL) challenged GES-1 cells (6 h). The release of IL-6 was assessed through an ELISA assay. The reference compound (black color) used was EGCG 20 μM (−24%). Data are reported as percentage in comparison to the stimulated control, which was arbitrarily assigned to 100% value. * *p* < 0.05, ** *p* < 0.01, *** *p* < 0.001 versus TNF-α. IL-6, interleukin-6.

**Figure 3 nutrients-14-01757-f003:**
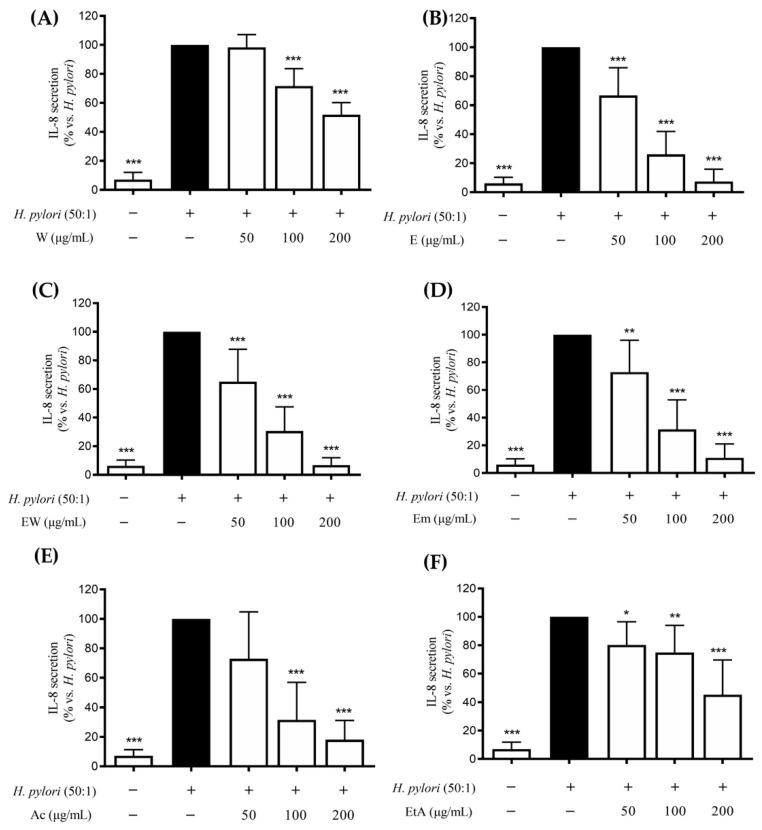
Effects of *Rhus coriaria* L. extracts ((**A**) 100% water (W), (**B**) 100% ethanol (E), (**C**) ethanol–water (ethanol:water 50:50 *v*/*v*, EW), (**D**) ethanol macerated (plant material subjected to maceration with pure ethanol for 48 h, Em), (**E**) acetone (Ac), (**F**) ethylacetate (EtA)) on the *H. pylori*-infected GES-1 cells (6 h). The cells were treated with *H. pylori*:cell ratio 50:1. The release of IL-8 was assessed through an ELISA assay. The reference compound (black color) used was EGCG 50 μM (−39%). Data are reported as percentage in comparison to the stimulated control, which was arbitrarily assigned to 100% value. * *p* < 0.05, ** *p* < 0.01, *** *p* < 0.001 versus *H. pylori*. *H. pylori*, *Helicobacter pylori*.

**Figure 4 nutrients-14-01757-f004:**
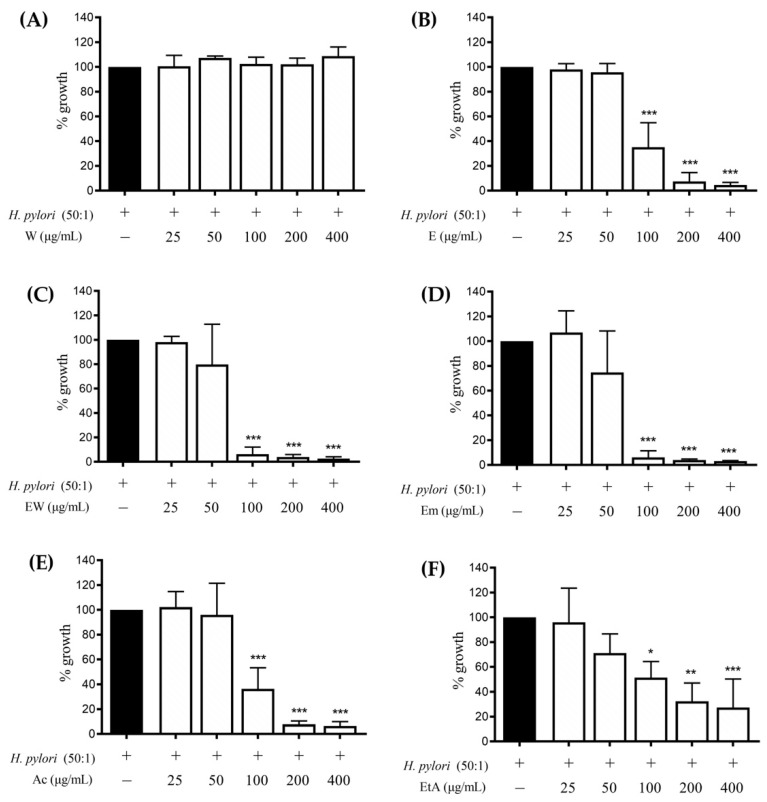
Effect of *Rhus coriaria* L. extracts ((**A**) 100% water (W), (**B**) 100% ethanol (E), (**C**) ethanol–water (ethanol:water 50:50 *v*/*v*, EW), (**D**) ethanol macerated (plant material subjected to maceration with pure ethanol for 48 h, Em), (**E**) acetone (Ac), (**F**) ethylacetate (EtA)) on *H. pylori* growth. Data are expressed in terms of growth inhibition of the bacterium compared to the control with only *H. pylori* (incubation time: 72 h), which was arbitrarily assigned value of 100%. The reference antibiotic (black color) used was tetracycline 0.125 μg/mL (100% inhibition). * *p* < 0.05; ** *p* < 0.01; *** *p* < 0.001 versus *H. pylori*.

**Figure 5 nutrients-14-01757-f005:**
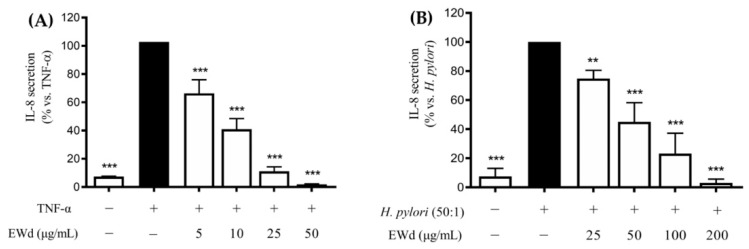
Effect of *Rhus coriaria* L. digested ethanol-water extract (EWd) on TNF-α (**A**) and *H. pylori*-induced IL-8 release in GES-1 cells (**B**). The release of IL-8 was assessed through an ELISA assay. EGCG 20 μM (−74%) was used as reference compound (black color). Data are reported as percentage in comparison to the stimulated control, which was arbitrarily assigned to 100% value. ** *p* < 0.01, *** *p* < 0.001 versus TNF-α or *H. pylori*. EWd, EW extract subjected to in vitro simulated gastric digestion.

**Figure 6 nutrients-14-01757-f006:**
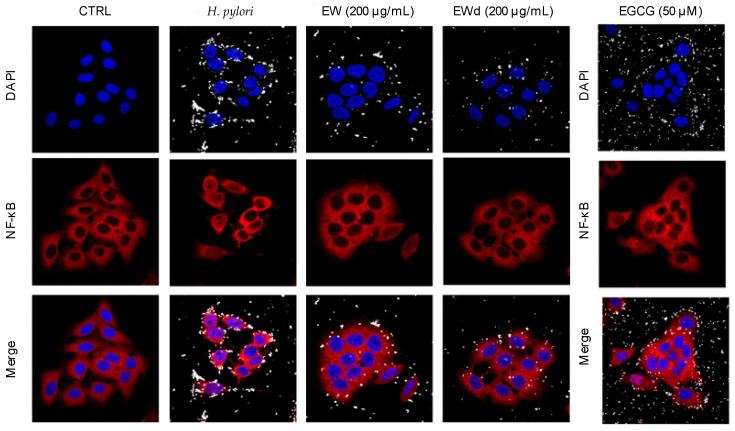
Immunofluorescence confocal microscope images of NF-κB p65 in GES-1 cells treated with the stimulus *H. pylori* (*H. pylori*:cell ratio 50:1) and the non-digested ethanol-water extract of *Rhus coriaria* L. (EW 200 μg/mL) or digested ethanol-water extract of *Rhus coriaria* L. (EWd 200 μg/mL) for 1 h. The reference compound used was EGCG 50 μM. Cell nuclei are represented in blue, the bacterium in white, and p65 subunit in red. NF-κB, nuclear factor kappa B; DAPI, 4′,6-diamidino-2-phenylindole; CTRL, control.

**Figure 7 nutrients-14-01757-f007:**
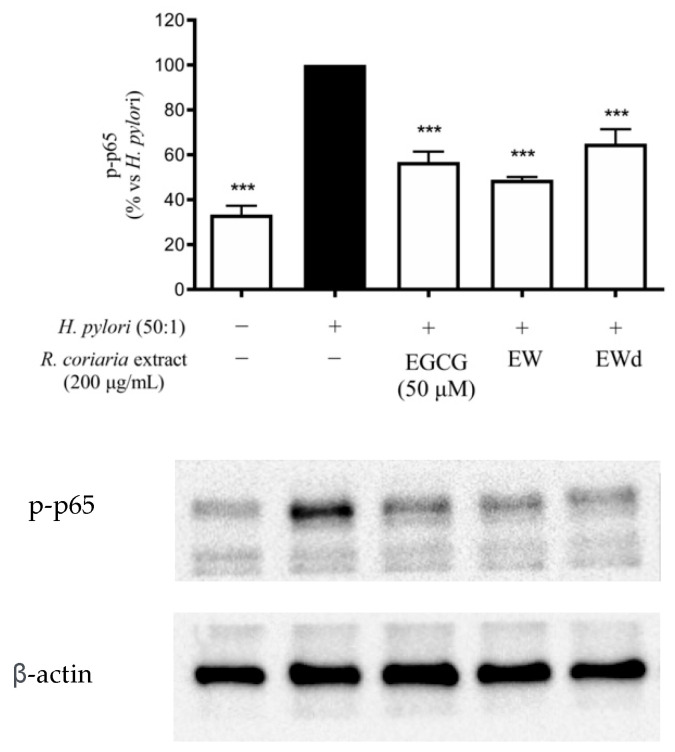
Effects of *Rhus coriaria* L. non-digested (EW) and digested (EWd) ethanol–water extracts on p-p65 activation in GES-1 cells treated with the stimulus *H. pylori* (*H. pylori*: cell ratio 50:1) for 1 h. The reference compound (black color) used was EGCG 50 μM (−43%). *** *p* < 0.001 versus *H. pylori*. p-p65, phospho-p65.

**Figure 8 nutrients-14-01757-f008:**
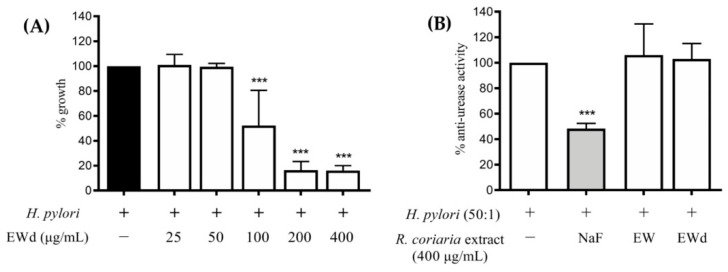
(**A**): effects of *Rhus coriaria* L. ethanol-water digested extract (EWd) on *H. pylori* growth. The data are expressed in terms of growth inhibition of the bacterium compared to the control with only *H. pylori* (incubation time: 72 h), which was arbitrarily assigned the value of 100%. The reference antibiotic used was tetracycline 0.125 μg/mL (100% inhibition). (**B**): anti-urease activity of *Rhus coriaria* L. ethanol-water not digested (EW) and digested (EWd) extracts compared with the reference compound (grey color) sodium fluoride (NaF) 1M (−52%). *** *p* < 0.001. *R. coriaria*, *Rhus coriaria*.

**Table 1 nutrients-14-01757-t001:** Total polyphenol content in *Rhus coriaria* L. extracts.

*Rhus coriaria* L. Extracts	Total Polyphenols (mg _Gallic Acid Equivalents_/g _extract_ ± S.D.)
Water (W)	93.2 ± 14.7
Ethanol (E)	224.7 ± 27.3
Ethanol-water (EW)	258.5 ± 37.1
Ethanol macerated (Em)	296.0 ± 39.0
Acetone (Ac)	223.6 ± 29.7
Ethylacetate (EtA)	82.6 ± 13.4

S.D., standard deviation.

**Table 2 nutrients-14-01757-t002:** Summary of IC_50_s (μg/mL) of *Rhus coriaria* L. extracts on IL-8 and IL-6 secretion induced by TNF-α in GES-1 cells.

*Rhus coriaria* L. Extracts	IL-8 Release (μg/mL)	95% C.I.	IL-6 Release (μg/mL)	95% C.I.
Water (W)	14.1	10.0 to 19.7	10.3	8.0 to 13.2
Ethanol (E)	14.7	11.0 to 19.7	19.3	13.1 to 28.5
Ethanol-water (EW)	12.7	8.8 to 18.5	19.3	11.8 to 31.6
Macerated ethanol (Em)	12.1	8.5 to 17.4	18.3	11.3 to 29.6
Acetone (Ac)	29.5	20.4 to 42.4	69.9	46.3 to 69.9
Ethylacetate (EtA)	32.5	21.8 to 48.4	85.4	66.6 to 109.5

IL-8, interleukin-8; IL-6, interleukin-6; TNF-α, tumor necrosis factor alpha; GES-1, gastric epithelial cells; C.I., confidence interval.

**Table 3 nutrients-14-01757-t003:** Summary of IC_50_s (µg/mL) of *Rhus coriaria* L. extracts on IL-8 and IL-6 secretion induced by *H. pylori* in GES-1 cells.

*Rhus coriaria* L. Extracts	IL-8 Release (μg/mL)	95% C.I.	IL-6 Release (μg/mL)	95% C.I.
Water (W)	183.1	150.9 to 222.1	>200	–
Ethanol (E)	62.0	52.2 to 73.7	>200	–
Ethanol-water (EW)	62.4	51.4 to 75.6	>200	–
Macerated ethanol (Em)	69.7	56.1 to 86.4	>200	–
Acetone (Ac)	70.1	49.2 to 100.0	>200	–
Ethylacetate (EtA)	166.2	109.3 to 252.8	>200	–

IC_50_s, half maximal inhibitory concentrations.

## Data Availability

Data supporting reported results are available at: corresponding author (Stefano Piazza, stefano.piazza@unimi.it) and last name (Enrico Sangiovanni, enrico.sangiovanni@unimi.it).

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
