# Peer review of "The Nutraceutical Properties of Sumac (Rhus coriaria L.) against Gastritis: Antibacterial and Anti-Inflammatory Activities in Gastric Epithelial Cells Infected with H. pylori"

_nutrients, 2022, doi:10.3390/nu14091757_

Round 1

Reviewer 1 Report

  1. Authors claim that "this is the first study assessing the anti-H. pylori potential of different Sumac extracts". Brief search revealed that direct anti-H. pylori activity of ethanol Sumac extract is described in https://academicjournals.org/journal/JMPR/article-full-text-pdf/CF7A69322180.pdf; anti-ulcer activity is described in https://www.researchgate.net/profile/Haqeeq-Ahmad/publication/344014415_Evaluation_of_Anti-ulcer_activity_of_hydro_alcoholic_extract_of_Post_Sumaq_Rhus_coriaria_Linn_in_Ethanol_induced_Gastric_ulcer_in_experimental_Rats/links/5f4dea7e299bf13c5073990a/Evaluation-of-Anti-ulcer-activity-of-hydro-alcoholic-extract-of-Post-Sumaq-Rhus-coriaria-Linn-in-Ethanol-induced-Gastric-ulcer-in-experimental-Rats.pdf. Please provide proper citations of these works and compare the results obtained.
  2. MIC of 100 ug/ml is relatively high to be of therapeutic value and unlikely to be reached using Sumac as a dietary supplement. Please provide relevant discussion on practical significance of your findings.
  3. "Citotoxicity" at line 147 is miss-spelled, please correct to "Cytotoxicity".
  4. Number of replicates should be indicated in figure captions.
  5. Authors provide IC50 values for IL-6 and IL-8 release without confidence intervals or standard errors. Moreover, I suggest to remove IC50s since 5-50 or 10-100 ug/ml range is too narrow for proper calculation of IC50; 4 orders of magnitude concentration range is generally considered to be adequate.
  6. Anti-inflammatory activity of extracts in CES-1 cells is much higher for TNF-alpha stimulation than for H. pylori stimulation. Please add discussion on this.

Author Response

Reviewer 1

1. Authors claim that "this is the first study assessing the anti-H. pylori potential of different Sumac extracts". Brief search revealed that direct anti-H. pylori activity of ethanol Sumac extract is described in https://academicjournals.org/journal/JMPR/article-full-text-pdf/CF7A69322180.pdf; anti-ulcer activity is described in https://www.researchgate.net/profile/Haqeeq-Ahmad/publication/344014415_Evaluation_of_Anti-ulcer_activity_of_hydro_alcoholic_extract_of_Post_Sumaq_Rhus_coriaria_Linn_in_Ethanol_induced_Gastric_ulcer_in_experimental_Rats/links/5f4dea7e299bf13c5073990a/Evaluation-of-Anti-ulcer-activity-of-hydro-alcoholic-extract-of-Post-Sumaq-Rhus-coriaria-Linn-in-Ethanol-induced-Gastric-ulcer-in-experimental-Rats.pdf. Please provide proper citations of these works and compare the results obtained.

We thank the reviewer for the suggested literature. We added the literature and commented the results in the “Discussion” section.

2. MIC of 100 ug/ml is relatively high to be of therapeutic value and unlikely to be reached using Sumac as a dietary supplement. Please provide relevant discussion on practical significance of your findings.

We added a specific comment in the discussion (line 472): according to the substantial maintenance the bioactivity of Sumac extracts (EWd) in the simulated gastric environment, it is plausible to suppose that the concentrations used in our experiments (ug/mL) may be achieved by dietary consumption of Sumac fruits or derived extracts (order of 10-100 grams of weight).

3. "Citotoxicity" at line 147 is miss-spelled, please correct to "Cytotoxicity".

Cytotoxicity was amended.

4. Number of replicates should be indicated in figure captions.

The number of replicates for each result (3 independent experimental replicates, at least) was specified in the Method section (line 262).

5. Authors provide IC50 values for IL-6 and IL-8 release without confidence intervals or standard errors. Moreover, I suggest to remove IC50s since 5-50 or 10-100 ug/ml range is too narrow for proper calculation of IC50; 4 orders of magnitude concentration range is generally considered to be adequate.

We thank the reviewer for the suggestion regarding statistical analysis. We added the confidence intervals in each Table regarding IC50s (Table 2, 3). In all the experiments in which an IC50 was calculated, the GraphPad software was able to draw the sigmoidal curve, which is the pharmacological parameter necessary to have reliable and robust IC50 values. The authors are confident that the parameter is reliable.

6. Anti-inflammatory activity of extracts in CES-1 cells is much higher for TNF-alpha stimulation than for H. pylori stimulation. Please add discussion on this.

We thank the reviewer for the prompt observation; we added a comment on this point in the Discussion section (line 451).

Reviewer 2 Report

Dear Authors

Thank you very much for your manuscript submission. Your study is well-designed and well-presented. However, some minor revisions are needed:

  1. In Materials and Methods section; Extract preparation and characterization subtitle: You have mentioned "All the extracts tested in this study were prepared according to [5]." It is better to complete your sentence as below:
    All the extracts tested in this study were prepared according to reference #5.
  2.  In Materials and Methods section; Bacterial culture subtitle: You have mentioned "H. pylori strain (ATCC American Type Culture Collection) was cultured in... " Please mention the # of ATCC.
  3. Please mention the related references for the used protocols in Materials and Methods section.
  4. Please read and add the following paper to the References section of the manuscript to represent more information regarding H.pylori in Introduction section:

    Advances in diagnosis and treatment of Helicobacter pylori infection. Acta Microbiol Immunol Hung. 2017 Sep 1;64(3):273-292. doi: 10.1556/030.64.2017.008. Epub 2017 Mar 6. PMID: 28263101.

Author Response

We thank the reviewer for positive comments. Please, find our point-by-point corrections below:

1. In Materials and Methods section; Extract preparation and characterization subtitle: You have mentioned "All the extracts tested in this study were prepared according to [5]." It is better to complete your sentence as below:
All the extracts tested in this study were prepared according to reference #5.

We thank the reviewer for the correction. The reference was amended.

2. In Materials and Methods section; Bacterial culture subtitle: You have mentioned "H. pylori strain (ATCC American Type Culture Collection) was cultured in... " Please mention the # of ATCC.

The strain definition was added in the Methods section.

3. Please mention the related references for the used protocols in Materials and Methods section.

References for the used protocols, including manufacturer instructions, were included.

4. Please read and add the following paper to the References section of the manuscript to represent more information regarding H.pylori in Introduction section:
Advances in diagnosis and treatment of Helicobacter pylori infection. Acta Microbiol Immunol Hung. 2017 Sep 1;64(3):273-292. doi: 10.1556/030.64.2017.008. Epub 2017 Mar 6. PMID: 28263101.

The reference was included in the introduction.